# Parenteral Cysteine Supplementation in Preterm Infants: One Size Does Not Fit All

**DOI:** 10.3390/biomedicines12010063

**Published:** 2023-12-27

**Authors:** Ibrahim Mohamed, Nadine El Raichani, Anne-Sophie Otis, Jean-Claude Lavoie

**Affiliations:** 1Department of Pediatrics/Neonatology, CHU Sainte-Justine, University of Montreal, Montreal, QC H3T 1J4, Canada; jean-claude.lavoie@umontreal.ca; 2Department of Nutrition, University of Montreal, Montreal, QC H3T 1J4, Canada; nadine.el.raichani@umontreal.ca; 3Pharmacy, CHU Sainte-Justine, University of Montreal, Montreal, QC H3T 1J4, Canada

**Keywords:** amino acid solution, antioxidant, cysteine, glutathione, parenteral nutrition, preterm infants, transmethylation, transsulfuration

## Abstract

Due to their gastrointestinal immaturity or the severity of their pathology, many neonates require parenteral nutrition (PN). An amino acid (AA) solution is an important part of PN. Cysteine is a key AA for protein and taurine synthesis, as well as for glutathione synthesis, which is a cornerstone of antioxidant defenses. As cysteine could be synthesized from methionine, it is considered a nonessential AA. However, many studies suggest that cysteine is a conditionally essential AA in preterm infants due to limitations in their capacity for cysteine synthesis from methionine and the immaturity of their cellular cysteine uptake. This critical review discusses the endogenous synthesis of cysteine, its main biological functions and whether cysteine is a conditionally essential AA. The clinical evidence evaluating the effectiveness of the current methods of cysteine supplementation, between 1967 and 2023, is then reviewed. The current understanding of cysteine metabolism is applied to explain why these methods were not proven effective. To respond to the urgent need for changing the current methods of parenteral cysteine supplementation, glutathione addition to PN is presented as an innovative alternative with promising results in an animal model. At the end of this review, future directions for research in this field are proposed.

## 1. Introduction

Term infants with specific pathologies and preterm infants are frequently unable to feed and need parenteral nutrition (PN) to provide adequate calories and necessary nutrients that promote growth and sustain essential biological functions. Amino acid (AA) solutions are an integral part of standard PN administered in neonatal intensive care units (NICUs). These AA solutions contain variable amounts of both essential and nonessential AAs. A low plasma level of any AA indicates a relative deficiency of this AA, which may be detrimental to nitrogen balance, growth and the specific functions related to this AA [1]. Multiple studies confirmed low plasma cysteine in parenterally fed neonates [2,3,4]. Cysteine plays an important role in glutathione (GSH) synthesis, which is a cornerstone in keeping the oxidant–antioxidant balance in preterm infants who are exposed to high oxidative stress induced mainly by O_2_ supplementation and PN contaminated with peroxide [5]. Multiple studies confirmed the presence of a significant amount of peroxide contamination in the PN [6,7]. The main source of these peroxides was found to be the multivitamins’ exposure to light that almost doubles the peroxide contamination through riboflavin photoexcitation [8,9,10]. A significant association between urinary peroxide concentration and death was reported in the adult population [11]. In premature infants less than 29 weeks gestational age, we reported similar results with a significant association between higher urinary ascorbyl peroxide (AscOOH) during the first week of life and increased incidence of bronchopulmonary dysplasia or death [12]. The intracellular availability of cysteine is essential to the effective glutathione system detoxification of these peroxides. This work is a critical review that applies the current understanding of cysteine metabolism (Section 1, Section 2, Section 3 and Section 4) to explain the results of the systematic review of clinical studies testing the effectiveness of clinically available methods of parenterally supplementing cysteine in neonates (Section 6) and presents an innovative promising alternative for effective parenteral cysteine supplementation in preterm infants (Section 7) [13].

## 2. Cysteine Sources

Cysteine is available through gut absorption from the diet [14]. In the case of infants on PN, some AA solutions contain cysteine, while others provide only methionine as a pro-cysteine AA. The different AA solutions and their content of cysteine and methionine are discussed in Section 4 of this review. The endogenous synthesis of cysteine with the sulfur atom of methionine and the carbon skeleton of serine makes it a dispensable amino acid under normal conditions [15]. As depicted in Figure 1, the first steps of cysteine synthesis are closely linked to the methionine cycle: methionine is converted by the action of the methionine adenosyltransferase into S-adenosylmethionine, which is subsequently converted to S-adenosylhomocysteine, and homocysteine. The following steps of cysteine synthesis take part in the transsulfuration pathway, where homocysteine is coupled to serine by the action of cystathionine-β-synthase, forming cystathionine, which will ultimately lead to cysteine through the cleavage action of γ-cystathionase [16].

## 3. Cysteine Functions and Biological Importance

### 3.1. Protein and Non-Protein Compound Synthesis

Like other amino acids, cysteine serves as a substrate for protein synthesis. Its unique thiol functional group among all amino acids allows for the formation of inter- or intrachain disulfide bonds between two cysteine molecules, through the oxidation of two thiols, leading to their conversion to one cystine molecule [14]. This specific functional group allows for cysteine to play a key role in protein structure and folding [17] and the redox regulation of several enzymes whose activities depend on the oxidation/reduction of their critical cysteinyl residue(s) [18,19]. This amino acid also serves as a substrate for non-protein compounds. Sulfate and taurine, for example, result from extensive oxidation of cysteine, while coenzyme A synthesis requires its incorporation [20].

### 3.2. Glutathione Synthesis

Cysteine plays a key role in the antioxidant defense, being one of the three amino acids forming glutathione (γ-glutamylcysteinylglycine), one of the most abundant antioxidants in the body [21]. The de novo synthesis of glutathione is essential in response to oxidant exposure. Intracellular synthesis is dependent on the intracellular availability of glutamate, cysteine and glycine [22] and is catalyzed by the two following enzymes: gamma-glutamate-cysteine ligase (γ-GCL) and glutathione synthase (GS). Whereas intracellular glutamate and glycine concentrations are higher than the Michaelis–Menten kinetics constant (Km) of the corresponding enzymes, the physiological cysteine concentration is close to the Km of the gamma-glutamate-cysteine ligase (γ-GCL) [23]. This makes the availability of cysteine a well-known limiting factor for glutathione synthesis [24,25]. This means that any change in the intracellular concentration of cysteine will have a direct impact on the enzymatic activity leading to glutathione formation. Because cysteinyl-tRNA synthetase has a greater affinity (Km = 1–20 µM [26]) for cysteine than γ-glutamylcysteine ligase (Km = 200–350 μM [27,28]), cysteine is preferentially used for the protein synthesis that is substantially important in premature newborns. Cells must therefore import sufficient cysteine to maintain protein and glutathione synthesis. 

Plasma glutathione levels in extremely preterm infants are 1.2 ± 0.1 μM (mean ± SEM) [12], while levels are 3–12 μM in full-term neonates [29,30]. Maternal venous and arterial umbilical cord blood cysteine and glutathione erythrocyte concentrations are significantly lower among preterm infants in comparison to healthy term ones [31]. This depletion in erythrocyte cysteine concentrations in very preterm infants at birth leads to glutathione depletion [31,32]. Thus, ensuring adequate intracellular cysteine availability is crucial for glutathione homeostasis [33]. This cysteine depletion leading to glutathione depletion has detrimental consequences in the context of prematurity birth that is associated with a high level of oxidative stress, resulting from the exposure to high levels of oxidizing molecules and low antioxidant defense capacities [34].

## 4. Is Cysteine a Conditionally Essential Amino Acid in Preterm Infants?

For the past decades, it has been unclear if cysteine is an essential amino acid for preterm infants [35]. To better understand the situations that can make cysteine a conditionally essential AA, this section describes the factors affecting cysteine bioavailability.

### 4.1. Factors Affecting Plasma Cysteine Concentration Subsection

Besides the nutritional intake of cysteine, there are several factors that can affect plasma cysteine. Clinically, these factors are associated with gestational age, postnatal age and the use of PN. The biological mechanisms can be resumed in the following two important steps of cysteine synthesis.

#### 4.1.1. Maturity of the Transsulfuration Pathway

The final step of the transsulfuration pathway leading to cysteine synthesis is catalyzed by the cleaving action of γ-cystathionase (Figure 1) [16]. Whereas some studies reported that the activity of this key enzyme is undetectable in the liver of fetuses or preterm infants that died during the first 24 h of life [36,37], others demonstrated that hepatic cystathionase activity in the first two weeks of life is reduced by half in preterm in comparison to full-term infants and appears to be positively correlated with gestational and postnatal age [38]. Similar findings were reported in animal studies; the hepatic cystathionase activity was lower in fetal and neonatal rats compared to adult rats [39]. The measurable activity of hepatic cystathionase in preterm infants [38] and the evident rates of transsulfuration among full-term enterally fed and preterm infants parenterally fed [40] may demonstrate the preterm capacity to synthesize cysteine but do not ensure that de novo cysteine synthesis is sufficient, particularly under major oxidative stress [41].

This lower hepatic cystathionase activity is associated with higher plasma cystathionine and lower cysteine concentrations in extremely and very preterm infants in comparison to moderate/late preterm and term infants [32]. The de novo synthesis of cysteine is dependent on the maturity of the transsulfuration pathway that is tightly linked to gestational age and postnatal age. Studies demonstrated that plasma cysteine concentrations on the first day of life are positively correlated with gestational age [42] or birth weight [34]. Whereas some authors reported no postnatal changes in plasma cysteine concentrations up to two weeks of life of preterm infants receiving PN, others reported rapid decrease in plasma cysteine in the first 12 hours of life that remained low thereafter [43,44], while others reported a significant increase in plasma and erythrocyte cysteine concentration on day 7 of life of extremely and very low birth weight infants receiving parenteral and some enteral nutrition [42].

#### 4.1.2. Activity of the Transmethylation Pathway

The first step of the transmethylation pathway and endogenous cysteine synthesis is catalyzed by the action of methionine adenosyltransferase, converting methionine into S-adenosylmethionine (Figure 1) [16]. This enzyme is a sensitive target of free radicals, and its activity is affected by oxygen, nitrogen radical species [45] and peroxides [46]. The gut immaturity of preterm infants requires the use of PN to ensure growth and survival [35]. This essential nutrition is, however, inherently contaminated with peroxides (due to the nutrients’ interaction with dissolved oxygen), causing a redox imbalance and deleterious consequences of oxidative stress in preterm infants [47]. PN peroxide contamination was shown to reduce methionine adenosyltransferase activity in newborn guinea pigs, leading to the more oxidized redox potential of glutathione [46].

### 4.2. Factors Affecting Intracellular Cysteine Availability

Besides the plasma cysteine concentration, another important factor that limits the intracellular cysteine availability is the maturity of the cellular cysteine uptake. Evidence of the sex and gestational age-dependent maturity of the cysteine uptake of cells in preterm infants suggests that even with a sufficient plasma cysteine concentration, some extremely preterm infants could have depleted intracellular cysteine [22]. 

In the light of these factors, it is expected that preterm infants less than 32 weeks of gestation on PN have the highest risk of cysteine depletion. This mechanistic comprehension can help with understanding the results of the randomized controlled trials conducted by Riedjik et al. concluding that, in very low birth weight infants with a gestational age < 29 weeks (*n* = 47) at one month of postnatal age (at 32 ± 0 and 36 ± 1 weeks post-menstrual age) tolerating full enteral feeds [48] and moderate preterm infants (gestational age 32–34 weeks) fully enterally fed (*n* = 25) [49] randomized to five different formulas only differing by cysteine concentrations, cysteine is not considered a conditionally essential amino acid. In these two studies, infants were ≥32 weeks PMA post menstrual age and fully enterally fed. Their cellular cysteine uptake, as well as the transsulfuration pathway, would be more mature. Furthermore, the oxidation-sensitive transmethylation pathway would not be affected as these infants were not receiving any PN. In these two studies, formula enrichment in cysteine ensured an exogenous supply of this amino acid [48,49]. Using the same mechanistic comprehension approach can help with understanding the results confirming that cysteine is a conditionally essential amino acid in preterm infants < 32 weeks due to decreased γ-cystathionase activity [32].

## 5. Different Available PN Solutions and Their Content of Cysteine and Methionine

The amino acid solutions available and developed for neonates include Travasol Blend C^®^ (Baxter corporation, Mississauga, ON, Canada), Primène^®^ (Baxter corporation, Mississauga, ON, Canada), AminosynPF^®^ (ICU Medical Canada Inc., St-Laurent, QC, Canada), Vaminolact^®^ (Fresenius Kabi, Hong Kong), TrophAmine^®^ (B. Braun Medical Inc., Bethlehem, PA, USA) and Premasol^®^ (Baxter corporation, Deerfield, IL, USA) [50,51,52,53,54,55]. 

Cysteine is unstable in solution and readily oxidizes to insoluble cystine [56,57]. For this reason, amino acid solutions contain little or no cysteine. Primene^®^ contains 189 mg of cysteine per 100 mL, whose bioavailability has not been demonstrated, and Vaminolact^®^ contains 100 mg of cysteine per 100 mL, whereas the other solutions do not contain cysteine [43,44,45,46,47,48]. The manufacturers of TrophAmine^®^ and AminosynPF^®^ suggest adding cysteine to their solutions when preparing parenteral feeds for newborns [51,54]. The addition of cysteine to the Trophamine^®^ and AminosynPF^®^ solutions, as recommended by the manufacturers of these solutions, improves the calcium and phosphorus solubility by lowering the pH of the solution [58]. Table 1 shows the cysteine and methionine content of the different available AA solutions.

In current practice, low cysteine intake can be compensated with moderate supplementation with methionine (if endogenous cysteine synthesis is not affected) [56]. Two other ways are also used: the addition of the hydrochloride form of cysteine, which is more soluble but may aggravate metabolic acidosis [59], and the addition of N-acetylcysteine (NAC), which is more soluble than cysteine but has limited bioavailability. NAC was suspected to be poorly deacetylated in preterm infants [60].

## 6. Are current Methods of Parenterally Supplementing Cysteine Effective and Why?

Over the last decades, many studies have been performed to determine whether parenterally supplementing cysteine or NAC to neonates will have a significant clinical impact. Given the well-documented low cysteine concentration in preterm infants [31,32], cysteine supplementation should result in positive nitrogen balance and improved growth, which was the primary outcome in many of these studies [61,62]. In addition, cysteine supplementation was expected to increase glutathione and reduce the incidence of neonatal oxidative stress-related pathologies (mainly BPD and ROP) [60]. In this section, we will review full-text published randomized, quasi-randomized or crossover studies that evaluated the effect of cysteine or N-acetyl cysteine parenteral supplementation on relevant biological outcomes including nitrogen balance and the glutathione system and on clinical outcomes including growth and oxidative stress-related diseases in preterm infants. 

Searches were conducted using MEDLINE (1967 to 2023) on the 10th of December 2023. The MeSH headings used were cysteine and parenteral nutrition. This search resulted in 213 results. Limiting the search to human studies only gave 151 results. When using the filter ‘newborn’, fifty-eight results were obtained. Screening the title of these papers looking for clinical trials comparing two different doses of cysteine identified a total of seven papers [61,62,63,64,65,66,67]. After reading the abstracts, two of these papers were excluded as they compare amino acid solutions with different content in methionine simultaneously with different contents of cysteine [63,67]. One paper was a Cochrane review [66]. While this Cochrane review was excluded, it helped in identifying other eligible studies. Of the six studies included in the Cochrane review, two studies were already identified in the four included studies [61,62], three studies included in the Cochrane review were excluded from our review as they were in abstract format only [68,69,70] and the last study was added to our results [60]. When searching the references of all the retained papers, one study was identified as eligible and was added to our review [71]. All the included studies are described in chronological order of their publication in Table 2 [60,61,62,64,65,71]. 

Overall, these studies indicate no significant improvement in neonatal growth or in oxidative stress-related diseases with parenteral cysteine or N-acetyl cysteine supplementation. Even when larger individual positive changes in GSH and total glutathione (reduced (GSH) form + disulfide (GSSG) form) were described with cysteine supplementation, this effect was not significant in smaller infants [64].

According to recent advances in our understanding of cysteine/GSH metabolism, these results were expected. In infants less than 32 weeks on exclusive PN as in Ahola et al. and te Braake et al. [60,71], the effectiveness of parenteral supplementation of cysteine is compromised by the immature cellular uptake of cysteine [22]. In addition, for Ahola et al.’s study [60], intravenous NAC was suspected to be poorly deacetylated in preterm infants. Meanwhile, in more mature infants, with the previously described maturation of the transsulfuration pathway with increasing gestational age [38], it could be expected that their improved ability to synthesize cysteine from the already available methionine in their PN could have rendered the supplementation with additional cysteine to have no added value as in the study of Courtney-Martin et al. [65].

The identified studies have multiple limitations that need to be addressed. All these studies, with the exception for Ahola et al.’s study [60], had a limited number of participants, with the groups ranging from 5 to a maximum of 21 infants with no adequately documented sample size calculation. While these small groups could be sufficient to determine plasma amino acid concentration changes and to a lesser extent GSH system changes, these small groups are not expected to have enough statistical power to detect changes in weight gain and oxidative stress-related outcomes. It is also important to note that the intervention in these studies was for a short period of time ranging from 3 to 7 days, with the exception of the study by Calkins et al. [64]. Many factors other than amino acid availability could interfere with neonatal growth, making the evaluation of neonatal growth over a short period of time less specific and more challenging. Another limitation of these studies is the population studied. While we know that cysteine cellular uptake and the transsulfuration pathway both mature with advancing gestational age and with a significant sex effect on the cellular cysteine uptake [22,38], most of these studies were not classified by gestational age nor the sex of the patients. 

For future studies aiming to increase intracellular cysteine availability, it will be recommended to include only infants <32 weeks gestational age on PN. This group has an immature transsulfuration pathway and decreased methionine adenosyltransferase activity with PN administration, making them the most vulnerable group to the lack of the pro-cysteine substrate in the PN as their endogenous synthesis of cysteine from methionine is largely compromised [38,46]. We also recommend using a cysteine substrate that is not dependent on the cellular uptake of cysteine, which is immature in this population [22]. A promising alternative for parenteral cysteine supplementation that meets this criterion is discussed in the following section of this review.

## 7. A New Promising Alternative for Parenteral Cysteine Supplementation

Due to its abundance in cells and its high cysteine content, which represents a third of its composition, glutathione (γ-glutamylcysteinylglycine) is considered as a cysteine reservoir for the organism [30,72]. Due to its γ-glutamyl group, glutathione cannot freely cross the cell membrane. Cells must therefore synthesize their own glutathione. In the 1970s and 1980s, the golden age of enzymology, Alton Meister pioneered the understanding of glutathione metabolism. Among other things, he proposed the ‘γ-glutamyl cycle’ [73,74]. Since then, the metabolism that supports this cycle has been used and cited frequently by scientists. Recently, in 2018, the metabolism of this cycle was revisited by Bachhawat AK and Yadav S. [75], who confirmed much of it while clarifying certain notions in the light of more recent knowledge. A summary of our current understanding of cysteine-glutathione metabolism is depicted in (Figure 2). 

The ectoenzyme γ-glutamyl transpeptidase (γ GT) is key in this cycle [76,77]. It hydrolyzes the gamma–peptide bond of glutathione, releasing the dipeptide cysteinylglycine, which is captured by the cell, where dipeptidases release the cysteine into the cytosol [78]. The natural substrates of γ GT are molecules containing a γ-glutamyl group, such as reduced glutathione (GSH), disulfide glutathione (GSSG) and glutathione-conjugated molecules [75,79,80]. The affinity of the enzyme for these forms of glutathione varies between 6 and 12 µM [79], whereas the normal concentration of glutathione in plasma is reported to be between 3 and 12 µM [29,30,72]. Unfortunately, the plasma concentration of glutathione in one-week-old premature infants is 1.2 ± 0.1 μM [12]. This low plasma glutathione can explain the low glutathione level measured in leukocytes of premature infants [81]. A cysteine deficiency in these infants has been demonstrated [2,3,4].

Knowing that γ GT, which matures in the first days of life of premature neonates [33], has a similar affinity for GSH and GSSG, our question was which form of glutathione is most suitable to act as a supplement to PN. Of course, GSH reactivity predicts interactions with other nutrients present in intravenous solutions. The better stability of GSSG, predicted by its chemical structure, was confirmed. After 24 h of incubation of the PN at room temperature, 11% glutathione remained in the GSH-supplemented PN solutions compared to 72% in the GSSG-supplemented solutions [82]. The 28% loss of glutathione in the GSSG-supplemented solution was explained by the disulfide exchange between the GSSG and cysteine present in the PN, generating the mixed disulfide cysteine-glutathione [83]. This latter molecule is also a substrate for γ GT providing cysteine to cells for de novo glutathione synthesis [84]. 

The GSSG supplementation of PN made it possible to prevent, in newborn guinea pigs, a deficiency in plasma and pulmonary glutathione [85,86]. By preventing pulmonary oxidative stress, this supplementation has also prevented the pulmonary loss of alveoli [33,37], which is a main characteristic of bronchopulmonary dysplasia observed in premature infants.

## 8. Future Directions

Considering the current understanding of cysteine-glutathione metabolism, its role in maintaining cysteine homeostasis and the results obtained in the animal model, it is essential to continue studying this alternative, particularly on the tissue levels of cysteine, to confirm the possibility of using GSSG as a pro-cysteine molecule in preterm infants requiring PN. Clinical studies in preterm infants are warranted to evaluate this alternative’s biological effects on the plasma cysteine level, growth, neonatal oxidative stress and important clinical outcomes, such as the incidence and severity of bronchopulmonary dysplasia.

## 9. Conclusions

Cysteine is an important AA in protein synthesis, growth and antioxidant functions as a limiting factor of glutathione synthesis. It is a nonessential amino acid under normal conditions, but it becomes an essential amino acid under specific circumstances, like prematurity, and under oxidative stress, such as receiving PN contaminated with peroxides. Immature cellular cysteine uptake from plasma in preterm infants is another important factor limiting its intracellular bioavailability. The current methods of supplementing cysteine were proven ineffective in clinical randomized controlled trials. Based on the presented evidence, there is an urgent need for changing the current methods of parenteral cysteine supplementation. Adding glutathione to PN is a promising alternative with positive results in the animal model and warrants evaluation in clinical settings.

## Figures and Tables

**Figure 1 biomedicines-12-00063-f001:**
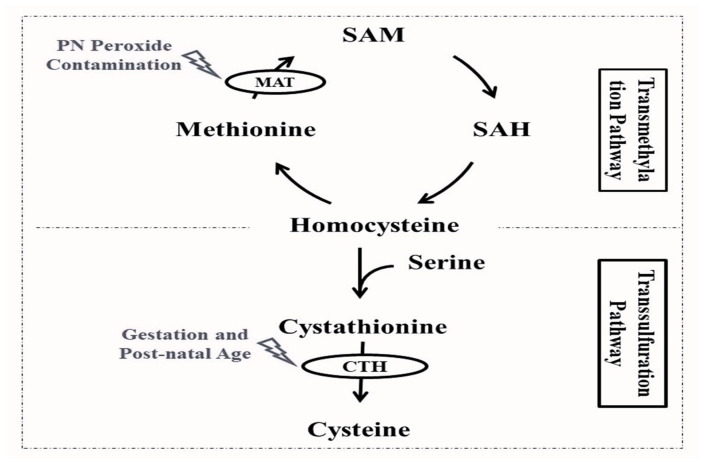
Cysteine endogenous synthesis through the transmethylation and transsulfuration pathways and affected enzyme activities among preterm infants. CTH: γ-cystathionase enzyme; MAT: methionine adenosyltransferase enzyme; PN: parenteral nutrition; SAH: S-adenosylhomocysteine; SAM: S-adenosylmethionine.

**Figure 2 biomedicines-12-00063-f002:**
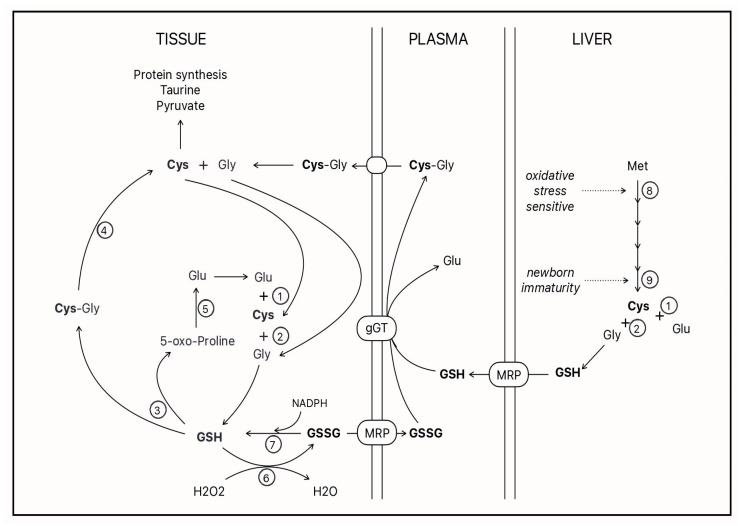
GSH: glutathione; GSSG: disulfide glutathione; Cys: cysteine; Gly: glycine; Glu: glutamate; Met: methionine; MRP: multidrug resistance protein; gGT: gamma-glutamyl transpeptidase. Open circles represent enzymes 1: gamma-glutamyl ligase; 2: GSH synthetase; 3: gamma-glutamyl cyclotransferase; 4: dipeptidases; 5: 5-oxo-prolinase; 6: glutathione peroxidase; 7: glutathione reductase; 8: methionine adenosyltransferase; and 9: cystathionase.

**Table 1 biomedicines-12-00063-t001:** L-cysteine and L-methionine contents in amino acid preparations [50,51,52,53,54,55].

	Travasol Blend C^®^	Primène^®^	Aminosyn PF^®^ 10%	Vaminolact^®^	TrophAmine^®^	Premasol^®^ 10%
**L-cysteine** (mg/100 mL)	0	189	0	100	<16	<16
**L-methionine** (mg/100 mL)	400	240	180	130	340	340

**Table 2 biomedicines-12-00063-t002:** Effect of parenteral cysteine/NAC supplementation on nitrogen balance, weight gain, glutathione and clinical outcomes.

Study	Patients	Intervention	Outcomes	Conclusion
Zlotkin et al., 1981 [61]	**Twenty-eight infants** were studied for a total of **36 episodes** of 6-day PN feeding periods. Seventeen of the infants were premature (gestational age 31 W (range 25 to 36)) and 11 were full term. Patients’ characteristics: **Inclusion criteria:** Infants received total intravenous nutrition for at least 6 days. **Exclusion criteria:** Infants receiving either plasma or blood transfusions were excluded. **Intervention** = 18 feeding episodes (gestational age 34.4 ± 2.2 W, birthweight 2231 ± 391 g, postnatal age 13.7 ± 5.5 d). **Control** = 18 feeding episodes (gestational age 34.2 ± 2 W, birthweight 2141 ± 357 g, postnatal age 15 ± 5.6 d).	**Quasi-randomized trial:** -Alternate assignment to intervention or control group. -All infants were NPO for at least 6 days. **Intervention:** Supplementation with an average of 78 ± 2 mg/kg/d cysteine for 6 days. **Control:** No cysteine supplement. There is no cysteine in the control amino acid formulation (Aminosyn^®^). **Study period:** Total of 6 days. The first 3 days were considered period of adaptation; all biological samples were collected during the last 3 days.	**Nitrogen balance:** Both groups showed similar positive retention of nitrogen, 282 mg/kg/24 h. **Plasma amino acid:** Plasma hemicystine was greater in the intervention group. No cysteine was found in the plasma samples. **Growth:** Weight change was not different between groups (control = 10.2 ± 1.7 g/kg/24 h; intervention = 6.1 ± 2.3). No difference in head circumference (control = 0.6 ± 0.1 cm/6 days; intervention = 0.9 ± 0.2) or length (control 0.6 ± 0.2 cm/6 days; intervention = 0.6 ± 0.1). **Long-term outcomes were not reported.**	Cysteine supplementation to cysteine-free parenteral nutrition failed to alter nitrogen retention and short-term growth.
Malloy et al., 1984 [62]	Twenty infants in 4 study groups (five infants in each group). **Inclusion criteria:** Two-day-old non-fed infants or infants who are unable to feed for at least 2 days and unlikely to start enteral feeding for at least 5–7 days. **Exclusion criteria:** Not provided. **Gestational age:** Not provided. **Birth weight:** Between 890 and 2500 g. **Days of life at enrollment:** Between 3 and 53 days.	**Randomized controlled trial:** -Infants are NPO and no intravenous lipid fat emulsion during the study period (6 days). -Intervention and control had low (240 mg/kg/d) and high (400 mg/kg/d) nitrogen intake subgroups. **Intervention:** Supplementation with 72 mg/kg/d cysteine added to PN for 6 days. **Control:** There is no cysteine in the control amino acid formulation (Aminosyn^®^).	**Nitrogen retention:** Was not affected by cysteine supplementation. **Plasma amino acids:** Cysteine supplementation had an effect only on free cyst [e] ine (mixture in any proportion of cysteine and cystine). **Weight gain:** Cysteine supplementation did not improve weight gain in the context of a low total daily non-protein caloric intake of 60–70 kcal/kg/d. **Long-term outcomes were not reported.**	No effect of cysteine supplementation on nitrogen retention nor the weight gain.
Ahola et al., 2003 [60]	Number of participants (*n* = 391) **Inclusion criteria:** On ventilator or nasal continuous positive airway pressure; enrollment before the age of 36 h. **Exclusion criteria:** Major congenital anomalies. **Birth weight:** Intervention 776 ± 128 g, control 780 ± 134 g. **Gestational age:** Intervention group (26.3 ± 1.7 W), control (26.5 ± 1.8 W).	**Randomized, double-blind, placebo-controlled, multi-center trial.** -Infants were on PN with some enteral nutrition. **Intervention:** NAC infusion was started before the age of 36 h and lasted for 6 days at a constant rate of 16 to 32 mg/kg/d. **Control:** Solvent without NAC. For both groups: Intravenous amino acids (Vaminolac).	**Primary outcome:** No difference in death by 36 gestational weeks or BPD. **Secondary outcomes:** No difference in the requirement of supplemental oxygen at the age of 28 days, duration of ventilator or nasal continuous positive airway pressure, weight gain, intraventricular hemorrhage, leukomalacia, necrotizing enterocolitis grade III or higher and retinopathy of prematurity (ROP). **Plasma amino acids:** There was a difference between the group means, but the increase in the plasma cysteine concentration from day 3 to day 7 was similar.	No beneficial effects of NAC with the dosage used could be demonstrated at 36 weeks corrected age.
te Braake et al., 2009 [71]	Twenty infants were enrolled in this study. **Inclusion criteria:** Very low birth weight infants (<1500 g). **Exclusion criteria:** Erythrocyte transfusions within 12 h before the study or during the study, and known congenital abnormalities; chromosomal defects; and metabolic, endocrine, renal or hepatic disorders. **Birth weight:** Intervention 1006 ± 120 g, control 978 ± 274 g. **Gestational age:** Intervention 27.4 ± 1.3 W, control 28 ± 1.7 W.	**Randomized controlled trial.** **Low dose group:** 45 mg/kg per day of cysteine (standard intake in Primene AA solution at an AA intake of 2.4 g/kg per day). **High dose group:** 81 mg/kg per day of cysteine (in Primene + additional cysteine). A stable isotope study to determine glutathione concentrations and synthesis rates in erythrocytes was conducted on day 2 of life.	**Glutathione concentration and synthesis rate in erythrocytes on postnatal day 2:** Glutathione concentrations and synthesis rates did not increase with additional cysteine administration. **Safety of cysteine supplementation** (reflected by base requirements in the first 6 days of life): Base requirements were higher in the high-dose cysteine group on days 3, 4 and 5. **Long-term outcomes were not reported.**	Administration of a high dose of cysteine (81 mg/kg/day) to preterm infants seems clinically safe but does not stimulate glutathione synthesis.
Courtney-Martin et al., 2010 [65]	Five neonates were enrolled in this study. **Inclusion criteria:** Infants born ≥ 34 wk gestation and ≤ 28 d chronological age at the time of the study. At least 3 d postoperatively. On PN providing adequate protein and calories. **Exclusion Criteria:** Chromosomal anomalies. Small for gestational age. Fever or documented infection. Presence of disease or on medications known to affect protein and AA metabolism. Receiving enteral feeds providing > 10% of protein intake. **Post-menstrual age:** 38.3 W. **Study weight:** 2400 g. **Postnatal age:** 1.8 W.	**Crossover study:** Each neonate was studied over 6 d. The study was divided in two parts. **In part 1 (3 days):** Infants received a parenteral AA solution devoid of cysteine, with the total sulfur amino acid (SAA) requirements provided as methionine only at the recommended dietary allowance (±56 mg/kg) (solution 1). **In part 2 (3 days):** Solution 1 + supplemental cysteine at 10 mg/kg (solution 2). On the third and the sixth days, a primed, continuous 7 h tracer infusion study was conducted to measure GSH kinetics.	**Plasma amino acids:** While cysteine concentration was higher with supplementation (173.7 ± 53.0 µmol/L versus 152.5 ± 43.5), this was not statistically significant. **GSH kinetics:** The additional 10 mg/kg/d of cysteine had no effect on GSH concentration, fractional or absolute synthesis rates. **Long-term outcomes were not reported.**	Results suggest that the total SAA requirement when provided as methionine only is adequate to meet the needs of the PN-fed human neonate for GSH synthesis and protein synthesis.
Calkins et al., 2016 [64]	Total of 48 subjects were enrolled, and 38 subjects (21 intervention group and 17 control) completed the study. **Inclusion criteria:** Neonates requiring mechanical ventilation who were <30 days of life with a score for neonatal acute physiology (SNAP) > 10 and projected PN requirement ≥ 7 days. **Exclusion criteria:** Renal or hepatic failure, insulin requirement, extracorporeal membrane oxygenation or a known inborn error of metabolism. Characteristics at infusion day: **Post-menstrual age:** Intervention (36 ± 5) W, and control (36 ± 6) W. **Weight:** Intervention (2500 ± 100) g, control (2400 ± 100) g. **Number of days of life at enrollment:** Intervention (14 ± 8) d, control (10 ± 4) d.	**Randomized controlled trial.** Stratified by birth weight (≤1499 g, 1500–2499 g, ≥2500 g). -All subjects received PN comprising dextrose, amino acids (10% Premasol^®^, Baxter, Deerfield, IL -containing minimal cysteine-HCl of 0.016 g/100 mL) and lipids (20% Intralipid^®^, Fresenius Kabi, Uppsala, Sweden). **Cysteine (CYS):** 121 mg/kg/day of supplement (approximately 40 mg/g-AA/day) cysteine-HCl. **Isonitrogenous non-cysteine supplementation (ISO):** 121 mg/kg/day of additional Premasol^®^ amino acids. -Supplement was continued until PN discontinuation or 60 days of PN, whichever came first. -Isotope study on PN day 7.	-The study was discontinued for slow recruitment after the preliminary analysis. -No significant difference in erythrocyte GSH, total glutathione concentrations, GSH/GSSG and FSR-GSH between groups. -Compared to the ISO group, the CYS group had a larger individual positive change in GSH and total glutathione (infusion day–baseline) (*p* = 0.02 for both). -Lower enrollment weight and red blood cell (RBC) transfusion were associated with a decreased change in total glutathione and GSH (*p* < 0.05 for both). -The BPD incidence, days on O_2_ and duration of hospital stay were not different between groups.	-High dose cysteine supplementation for at least one week in critically ill neonates resulted in a larger and more positive individual change in glutathione. -Smaller infants and those who received transfused blood demonstrated less effective change in glutathione with cysteine supplementation.

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
