# Peer review of "Parenteral Cysteine Supplementation in Preterm Infants: One Size Does Not Fit All"

_biomedicines, 2023, doi:10.3390/biomedicines12010063_

Round 1
Reviewer 1 Report
Comments and Suggestions for Authors
The review provides a concise overview of the subject, effectively introducing the importance of cysteine in parenteral nutrition for neonates, especially preterm infants. It clearly outlines the main topics covered, such as the endogenous synthesis of cysteine, its biological functions, and the consideration of cysteine as a conditionally essential amino acid. The mention of clinical evidence and the exploration of alternative supplementation methods, like glutathione, adds a future perspective to the review's scope. The inclusion of future research directions is a positive aspect that encourages continued exploration in the field. The article is sufficiently well-structured, with a logical flow of information from the background on cysteine's importance to the exploration of available amino acid solutions and clinical evidence. Each section is introduced and concluded effectively, aiding in the reader's understanding. The inclusion of information on available amino acid solutions and their cysteine and methionine content demonstrates a comprehensive understanding of the topic.
Despite all these positive aspects, there are some major concerns that should be addressed by the authors, in order to improve the quality of the review.
The article frequently employs non-academic vocabulary, which could impede clarity and hinder the understanding of the content for a specialized readership. Substituting some of the non-academic terms with more technical or precise language might enhance the overall professionalism and accessibility of the manuscript.
In addition, the manuscript contains numerous writing errors that range from grammatical issues to sentence structure problems. These errors can detract from the overall readability of the article. A thorough proofreading and editing process should be undertaken to rectify these issues, ensuring a polished and professional final structure.
One notable gap in the article is the absence of data regarding peroxides contamination in parenteral nutrition formulations. Given the critical nature of this aspect in the context of neonatal parenteral nutrition, the omission of such data leaves a significant void in the completeness of the review. Including relevant information on peroxides contamination would contribute to a more comprehensive understanding of the challenges and considerations in this field. Moreover, the presence of such impurities in PN formulations is a major concern from the pharmaceutical control perspective.
The conclusion of the article falls short in providing a clear and sharp stance on the immediate need for cysteine supplementation in neonatal parenteral nutrition. It is crucial to emphasize, based on the evidence presented, whether the current practices are sufficient or if there is an urgent requirement for changes in cysteine supplementation protocols. A more decisive conclusion would help readers understand the practical implications of the discussed research and guide potential adjustments in clinical practices.
The article could benefit from incorporating a more explicit personal clinical perspective from the authors. Including insights into the authors' viewpoints and experience regarding this research subject could add depth and nuance to the narrative. This could foster a stronger connection between the authors and the health professionals, enhancing the overall engagement with the topic.
Addressing all these comments and observations could increase the overall importance and impact of the review.
Comments on the Quality of English LanguageThe article frequently employs non-academic vocabulary, which could impede clarity and hinder the understanding of the content for a specialized readership. Substituting some of the non-academic terms with more technical or precise language might enhance the overall professionalism and accessibility of the manuscript.
In addition, the manuscript contains numerous writing errors that range from grammatical issues to sentence structure problems. These errors can detract from the overall readability of the article. A thorough proofreading and editing process should be undertaken to rectify these issues, ensuring a polished and professional final structure.
Reviewer 2 Report
Comments and Suggestions for Authors
Parenteral Cysteine Supplementation in Neonates: One Size Does Not Fit All" (biomedicines-2738452), submitted for the special issue "Advanced Research in Neonatal Pharmacology", aims to discuss the endogenous synthesis of cysteine, its main biological functions, and the potential of cysteine being a conditionally essential amino acid under certain circumstances.
Comments for Article Improvement:
- Article Summary: It is suggested that the summary should not only present the objective and rationale of the study, but also describe the design of the review conducted, the period it covers, and the data sources used. This would facilitate linking this review with future research on the subject. Additionally, it would be beneficial to present the main findings of the review, providing readers with an overview of the work's content. Therefore, it is recommended to rewrite the summary to include these elements.
- Keywords and Title: It is proposed to add "in neonates" to the keywords or to modify the title to include "preterm infants", thus ensuring coordination between the title and the keywords.
- Data Table Update: Table 2, though chronologically organized, is notable for its last entry in 2016. Using the PubMed database and the keywords "parenteral cysteine neonates", 189 articles on this topic were found for the period 2017 to 2023. Therefore, it is deemed necessary to update and complete the information presented in the table.
- Review Methodology: It would be useful to specify what type of review has been conducted and the methodology applied to aid in understanding the review. Although it appears to be a comprehensive review, clarity in this aspect is crucial.
- Bibliography Update and Methodology Incorporation: It is essential to update the bibliography and add details about the methodology used in the study to maintain the relevance and accuracy of the research.
These proposed changes will enhance the quality of the article, making it more complete, informative, and useful for readers interested in neonatal pharmacology.
Round 2
Reviewer 1 Report
Comments and Suggestions for Authors
The article has been substantially improved, it is now ready for publication.
Reviewer 2 Report
Comments and Suggestions for Authors
I have thoroughly reviewed the latest version of the review article titled "Parenteral Cysteine Supplementation in Neonates: One Size Does Not Fit All" (biomedicines-2738452), as well as the authors' responses to the suggestions made. I believe the article has been substantially improved, incorporating all the suggested points in detail. Furthermore, I would also like to highlight the input provided by the other reviewer, which has enhanced the quality and clarity of the text and the message of the work undertaken.